# Improving Syrian refugees' knowledge of medications and adherence following a randomized control trial assessing the effect of a medication management review service

Majdoleen Alalawneh[1], Alberto Berardi[2], Nabeel Nuaimi[3], Iman A. Basheti[4,5]*

1 Department of Pharmaceutical Sciences, College of Pharmacy, QU Health, Qatar University, Doha, Qatar, 2 Department of Pharmaceutical Sciences and Pharmaceutics, Faculty of Pharmacy, Applied Science Private University, Amman, Jordan, 3 Department of Clinical Pharmacy, College of Pharmacy, AlNoor University College, Mosul, Iraq, 4 Department of Clinical Pharmacy and Therapeutics, Faculty of Pharmacy, Applied Sciences Private University, Amman, Jordan, 5 School of Pharmacy, The University of Sydney, Sydney, Australia

* dr_iman@asu.edu.jo

**Data Availability Statement:** All relevant data are within the paper and its Supporting Information files.

## Abstract

### Background

Syrian refugees living in Jordan have many chronic conditions and use many medications. Pharmacists delivering the Medication Management Review (MMR) service can have a role in improving this growing global refugees' problem.

### Objectives

To assess the effect of the MMR service on adherence to treatment therapy and knowledge of chronic medications for Syrian refugees residing in Jordan.

### Methods

This randomized intervention control single-blinded study was conducted in Jordan. Syrian refugees were recruited and randomized into intervention and control groups. Two home visits were delivered to each participant, at baseline and three months later. All participants completed questionnaires regarding adherence and knowledge. As a part of the MMR service, treatment-related problems (TRPs) were recognized for all patients; recommendations to resolve these TRPs were only delivered to intervention group refugees' physicians; TRPs were corrected. At follow-up, TRPs assessment, adherence and medication knowledge were assessed for all refugee participants.

### Results

Participants (n = 106; intervention n = 53, control n = 53) had a number of medications and diagnosed chronic diseases of 5.8 ± 2.1 and 2.97 ± 1.16 per participant respectively. A significant improvement in the adherence and knowledge scores were noted in the intervention (P < 0.001 for both) but not the control group (P = 0.229, P = 0.07 respectively).

**Funding:** The author(s) received no specific funding for this work.

**Competing interests:** The authors have declared that no competing interests exist.

## Conclusion

The MMR service can significantly improve refugees' TRPs, adherence to therapy and knowledge of chronic medications. If this approach was extended to the large scale, many refugees in need would be able to access a quality essential health-care service; a step towards achieving universal health coverage.

## Trial registration

**Registry:** ClinicalTrials.gov Identifier: NCT04554810.

## Introduction

The United Nations reported that, as of February 2016, a total of 13.5 million Syrian refugees had been identified as in need of humanitarian aid; of these refugees, 6.6 million were displaced within Syria and 4.8 million outside of Syria [1]. Refugees in general experience low living conditions, limited access to health care services as well as high poverty level in the country of settlement [1]. Taking these factors into consideration, refugees often have multiple health issues that need extensive health care [1–6]. In this context, low adherence to treatment and poor knowledge of medications and therapeutic regimens can be expected. Adherence to treatment can be an issue to many refugees, as getting their medications on time, and being provided with enough education on the importance of taking their medications as prescribed can be lacking [7, 8]. Medication knowledge has been defined as 'the awareness of the drug name, purpose, administration schedule, adverse effects or side-effects, or special administration instructions' [9]. It has been found that patients who need to take various medications are usually deficient in medication knowledge [10].

The term treatment related problem (TRP) is widely used in the literature and is approved by major clinical areas across the world [11, 12]. The TRP can be defined as "an event or circumstance involving patient treatment that actually or potentially interferes with an optimum outcome for a specific patient". It has been demonstrated in a previous longitudinal German study conducted between 2003 and 2007 that more than 5% of hospital visits are either caused or complicated by TRPs [13]. Locally, a Jordanian study revealed that the identified TRPs in Jordanian outpatients with chronic diseases visiting community pharmacies are high [14].

Globally, pharmaceutical care services have improved the number of TRPs among certain population, the use of medications, and patients' adherence to their treatment [15]. In particular, patients' outcomes have been improved in various countries and in different populations and settings, through the use of the pharmacist-led Medication Management Review (MMR) service [16, 17]. This service is defined as "a distinct service or group of services that optimize clinical outcomes for each participant to ensure the appropriateness, effectiveness and safety for each participant's medication(s); in addition to ensuring the ability of the participant to take his/her medication(s) as should be" [18]. MMR services are designed to improve education of chronically ill patients, based on a comprehensive review of their treatment, past and current situation, in order to ultimately resolve adherence and medication knowledge issues [19, 20]. With the growing crises in Syria since 2011, large numbers of Syrian people have fled to other countries, including Jordan. The current "refugee crisis" has escalated sharply in Jordan and its impact on health and financial sections are widening [21]. To overcome this hypercritical situation, many camps were established in the country, such as Za'atari, Marjeeb al-

Fahood, Cyber City and Al-Azraq camps. The UNHCR's Home Visits Program Report has estimated that the largest number of Syrian refugees in Jordan are living outside these camps (84%); while the remaining are settled in urban and rural areas [22, 23]. Primarily, Syrian refugees do not have a work permit to work in Jordan, and they are employed in the informal economy and they are not covered in the bounds of Jordanian labour law. However, approximately 51% from the Syrian refugees who lives outside camps participates in the Jordanian labour market to afford an income for their families [24]. The unemployment rate for the total Syrian refugees in Jordan is considered high and it reaches 57% [24]. Consequently, lower income, harder working conditions, tighter regulations, and lack of good work contracts are among the features of such "work" for the refugees [24].

In this scenario, Syrian refugees are facing many health-related problems, as their health care needs are not completely fulfilled [21]. Unfortunately, healthcare needs of Syrian people, based on their reported situation, remain still to be answered; thus, it is important to evaluate, and possibly address, refugees' healthcare needs in the countries where they settled in. No previous study, in Jordan or abroad, has assessed the needs and impact of the MMR service on refugees' adherence to their treatment therapy, and knowledge of their treatment.

The primary aim of this study was to assess refugees' adherence and knowledge of their chronic medications, and impact of the MMR service delivered by a clinical pharmacist on their adherence and knowledge of their chronic medications three months following delivery of the service.

## Method

### Study design and clinical setting

This study was a randomized interventional clinical study, conducted over six months (May to October 2016) in different Jordanian cities, where most of Syrian refugees reside. The registration of this clinical study delayed, as it is not requested by Jordan, and due to time constraints (ClinicalTrials.gov Identifier: NCT04554810); The authors confirm that all ongoing and related trials for this drug/intervention are registered. Ethics approval was obtained from the Jordanian Ministry of Health (MOH REC 160079).

Clinics which are specialized for Syrian refugees were approached by the clinical pharmacist (researcher) in order to meet Syrian refugees, recruit eligible participants and arrange for their first home visits. One clinical pharmacist, who has four years' experience in the field (with no additional qualifications), has recruited the Syrian refugees and has delivered the MMR service to them.

Inclusion criteria for the Syrian refugees included $\geq$ 18 years, living in Jordan for more than six months prior to study recruitment and intending to stay for the whole study period, having one chronic condition at least or taking 5 or more medications or taking more than 12 doses of a medication per day [25]. There was a deviation from the original study protocol regarding the sample size, as the calculated needed number of patients was 138 patients, however the recruited number was 109, due to time restriction.

An informed consent form was signed by all participants who accepted to participate. Eligible participants were then randomized (sequence-generation randomization) into intervention and control groups using a predetermined list obtained by the computer randomization program (www.randomizer.org) before starting the study. The study was single–blinded, hence, participants were not informed of the group they were randomized into, but they were informed that they would have been in either of two study groups. The first group would have received the MMR service during the study period, while the to the other group directly after

the study was completed (after three months' time). Random sequencing, participants' enrolment, and participants' assignment to interventions were done by the clinical pharmacist.

**Sample size.**   Sample size calculation was based on the primary outcome variable, adherence to treatment, improvement before and following involvement in the MMR service. Depending on previous studies [19, 26], in order to detect a significant amendment in TRPs of 1 point difference [26], with a power of 85%, significance level of 5% and standard deviation of the change of 2.09 (variance based on the data from a previously published study [26]. The sample size of 138 was the needed required sample in both groups.

## Study protocol

After the eligible patients were randomized into two groups, intervention group and control group, appointments were arranged at the physicians' clinics for all participants to be visited by the clinical pharmacist at their homes. At the baseline home visit, the clinical pharmacist documented participants' demographics, acute and chronic medical problems, history of present diseases, past medical history, lifestyle, family history, allergies, vital signs, physical examination information, diagnostic test data, lab results, current medications and issues related to the short and long term management of the patients from both groups.

The MMR service was completed following verification of collected baseline data. The home visits were planned not to exceed one hour, which is the usual time provided in related studies [27]. During these visits, self-completed questionnaires were completed by the participants from both groups, evaluating their adherence and knowledge of their chronic medications.

The clinical pharmacist identified and documented the treatment-related problems (TRPs) for each patient in both groups at baseline. All identified TRPs were supported by current therapeutic guidelines, reported in a letter format sent to the participant's physician. In the case any life-threatening TRPs were identified for control groups participants, they were excluded from the study for ethical considerations.

The physician was identified based on the participant's reported clinic and on participant's choice when more than one physician was visited by the participant. Following receipt of the letter, physicians addressed the recommendations and returned the letter to the pharmacist showing approved and rejected recommendations. Participants from both groups were called by the pharmacist to visit the physician and have the approved recommendations applied. Counselling and education were delivered to participants in the intervention group regarding their illnesses, knowledge of medications and adherence to their treatment.

**Follow-up assessment.**   Three months post baseline, new appointments were arranged through a phone call by the clinical pharmacist, and all participants were revisited at home. Data needed to assess TRPs were recollected (as was done at baseline), plus the adherence and knowledge of chronic medications' questionnaires was completed for all participants. At the end of the study, control group participants received the MMR and pharmacist counselling service exactly as was delivered to the intervention group participants at baseline [28].

## Data collection tools

The primary outcome was measuring the impact of the MMR service on the adherence to medications, and knowledge about drug therapy among the patients. For the purpose of data documentation and evaluation, the following self-completed questionnaires were used. Patients were asked to answer these questionnaires by themselves. However, in case of a certain patient was unable to read/understand the given questionnaire, the appropriate help was provided by the clinical pharmacist.

## Adherence to medication

Adherence to medications was assessed by using a questionnaire which was developed, validated and published by AbuRuz et al. [29], based on a scale developed by Morisky et al. [30].

The questionnaire was composed of eight questions, the first 5 questions are as follow: how often the patient during the last month forgot to take his/her medication/s, skipped it, stopped it when feeling better, stopped it when feeling worse or stopped it when they experienced a side effect. The measurement scale used in this questionnaire was scored at 0 (never), 1 (rarely), 2 (sometimes), 3 (often) and 4 (always). Hence, adherence was analysed as a likert scale out of 5. Higher scores indicted lower adherence by the patient. The patients were considered under-adherent if they scored 1.0 or more in the total [30].

In the sixth and seventh questions, patients' limit of commitment to pharmacist advice and their main cause of non-adherence were also assessed. Lastly, patients were asked about the main reason behind their non-adherence to their medications in the eight question of the questionnaire. The possible listed reasons for non-adherence were as follow: price, timing of dose, forgetting, I do not like medicines, medicine does not work, side effects, high number of pills every day.

For the intervention group, the clinical pharmacist answered any queries raised by the patients following their adherence assessment at baseline and study follow-up. However, for the control group, the clinical pharmacist did not answer any queries made by the patients following assessment, but for ethical reasons, all queries were recorded and answered after the end of the study.

## Knowledge of chronic medications

Knowledge of chronic medications was evaluated using an ad hoc questionnaire, based on a previously validated and published one (Knowledge about Drug Therapy questionnaire) [19, 29, 31]. The ad hoc questionnaire was used in this study for the purpose of facilitating the reading and understanding by Syrian refugees. The ad hoc questionnaire consisted of five questions related to patients' medications including 1- scientific medication name, 2- generic medication name, 3- how, 4- when, and 5- why was the patient taking each medication. Medication knowledge was assessed as an aggregate measure for all medications per patient.

The measurement scale used in this questionnaire was scored at 0 (I know this very well), 1 (I know this to some extent), 2 (I know for some of my medications only), 3 (I do not know) and 4 (I do not know at all). Hence, knowledge of chronic medication was analysed as a Likert scale out of 5. Higher scores indicated less knowledge of one's medication therapy.

## Statistical analysis

Data were coded then entered into the Statistical Package for Social Sciences (SPSS), version 20. Continuous variables were expressed as mean ± SD. Differences within the same group were detected using paired sample t-test for continuous variables. Categorical data were expressed as proportion (%) and analysed using Chi-square test. Likert scale scores (for adherence and knowledge of medication questionnaire) were analysed using the generalized linear model ordinal logistic regression. A probability value of $< 0.05$ was considered to be statistically significant for all analysis's tests.

## Results

A total of 123 Syrian refugee participants was approached for recruitment into the study; the recruitment was carried out over three months (i.e., May to July 2016). After participants' first

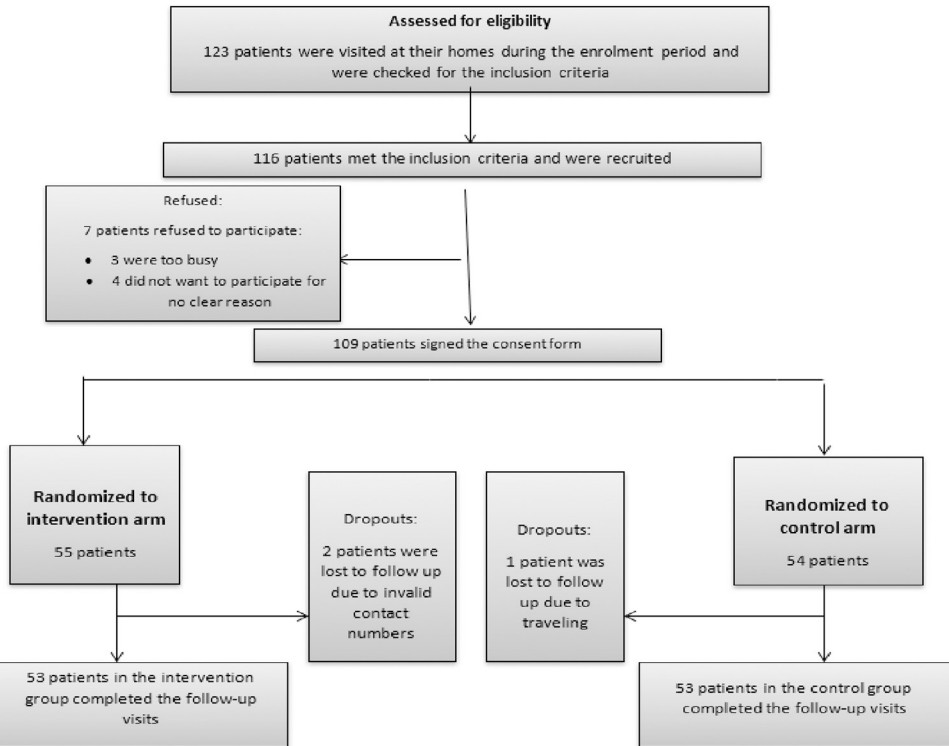

**Fig 1. Consort diagram showing patients' recruitment and retention during the study period.**

meeting during the home visit by the clinical pharmacist, 116 participants (94.3%) were found eligible for study and 7 participants refused to continue in the study due to different causes (Fig 1). Participants (n = 109) were randomized into intervention (n = 55) and control (n = 54) groups thereafter. The acceptance rate for home visits was 100%. No participants with life-threatening cases were identified in the control group. Follow-up visits were completed over three months (i.e., August to October 2016).

After dropouts (3 patients), all participants (n = 106, 53 in each group) completed the study (data analysis was completed for the 106 participants). Mean age of the participant was 58.5 ± 11.2, with about half being females (n = 51, 48.1%) and 51.9% had primary or preparatory educational level (Table 1). The approval rate of pharmacist's recommendations by the contacted physicians was high (82.9%) [28].

The number (mean ± SD) of medications used per patient was 5.8 ± 2.1, and the number of chronic conditions per participant was 2.97 ± 1.16. There were no significant differences between the two groups' participants with regards to the number of medications (P = 0.099) and the number of chronic conditions (P = 0.054). Diabetes and hypertension were the most common chronic conditions diagnosed among the participants (66.0%), followed by dyslipidemia (44.0%), cardiac illness (39.0%) and asthma (9.0%). The outcomes regarding the impact of the MMR on the TRPs' type and frequency are reported elsewhere [28].

## Adherence to medications at baseline and follow-up

A significant improvement in the adherence scores was noted in the intervention group across the study. No significant improvement was found in the control group regarding the mean change in the adherence score (Fig 2). From the factors which were assessed (number of

**Table 1. Demographic characteristics of the study sample.**

| Parameter | Total number of patients | Intervention group | Control group | P-value |
|---|---|---|---|---|
| **Number of patients, n (%)** | 106 (100) | 53 (50) | 53 (50) | -------- |
| **Age** (year), **mean (SD)** | 58.5 (11.2) | 60.1 (12.1) | 56.9 (10.1) | 0.157[1] |
| **Gender n (%)** | | | | |
| Male | 55 (51.9) | 26 (49.1) | 29 (54.7) | 0.560[2] |
| Female | 51 (48.1) | 27 (50.9) | 24 (45.3) | |
| **BMI mean (SD)** | 30.24 (7.1) | 30.3 (7.02) | 30.2 (7.3) | 0.813[3] |
| **Body mass category,(n) %** | | | | |
| Under weight | 4 (3.8) | 3 (5.7) | 1 (1.9) | 0.469[2] |
| Normal | 16 (15.1) | 9 (17.0) | 7 (13.2) | |
| Over weight | 32 (30.2) | 12 (22.6) | 20 (37.7) | |
| Obese class 1 | 36 (33.9) | 18 (34) | 18 (34) | |
| Obese class 2 | 12 (11.3) | 8 (15.1) | 4 (7.5) | |
| Morbid obesity | 6 (5.7) | 3 (5.6) | 3 (5.7) | |
| **Marital status, n (%)** | | | | |
| Single | 3 (2.8) | 2 (3.8) | 1 (1.9) | 0.488[2] |
| Married | 83 (78.3) | 39 (73.6) | 44 (83.0) | |
| Widow | 20 (18.9) | 12 (22.6) | 8 (15.1) | |
| **Exercise, n (%)** | | | | |
| Yes | 30 (28.3) | 13 (24.5) | 17 (32.1) | 0.388[2] |
| No | 76 (71.7) | 40 (75.5) | 36 (67.9) | |
| **Smoking, n (%)** | | | | |
| Yes | 44 (41.5) | 20 (37.7) | 24 (45.3) | 0.430[2] |
| No | 62 (58.5) | 33 (62.3) | 29 (54.7) | |
| **Caffeine, n (%)** | | | | |
| Yes | 81 (76.4) | 41 (77.4) | 40 (75.5) | 0.819[2] |
| No | 25 (23.6) | 12 (22.6) | 13 (24.5) | |
| **Education level, n (%)** | | | | |
| Primary + preparatory | 55 (51.9) | 30 (56.6) | 25 (47.2) | 0.299[2] |
| High school | 29 (27.4) | 16 (30.2) | 13 (24.5) | |
| BSc | 19 (17.9) | 6 (11.3) | 13 (24.5) | |
| MSc | 3 (2.8) | 1 (1.9) | 2 (3.8) | |
| **Clinic, n (%)** | | | | |
| Emirate Red Crescent | 40 (37.7) | 20 (37.7) | 20 (37.7) | 0.742[2] |
| Jordan Health AID Society | 32 (30.2) | 14 (26.4) | 18 (34.0) | |
| Private | 30 (28.3) | 16 (30.2) | 14 (26.4) | |
| Social charity | 3 (2.8) | 2 (3.8) | 1 (1.9) | |
| Governmental center | 1 (0.9) | 1 (1.9) | 0 (0.0) | |

**n**: Number of patients

**SD**: Standard deviation

**1** Analysis by independent-sample t-test

**2** Analysis by chi-square test

**3** Analysis by Mann Whitney U-te

medications, number of medical conditions, age, and education level), the number of medications was the only factor which affected the change in adherence scores between baseline and follow-up among patients (P = 0.009).

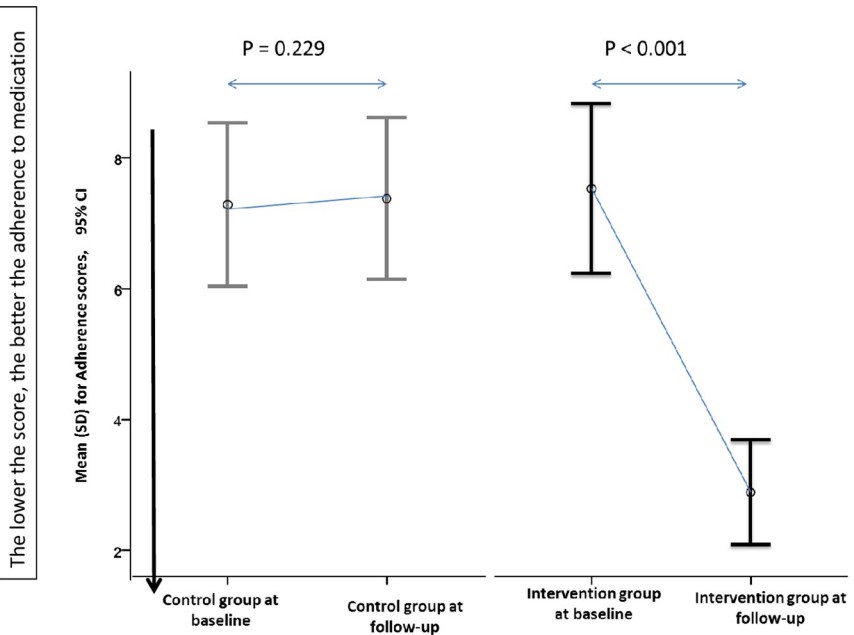

**Fig 2. 'Adherence to medication' mean changes at baseline and follow-up.**

With regards to the proportions of adherent patients, there was no significant difference between the intervention and the control groups at baseline (P = 0.54, Chi-square test) where 13.2% of patients in the intervention group were adherent to medications versus 11.3% in the control group. Due to receiving the MMR service, a significant difference between the two groups was found at three months (P = 0.01, Chi-square test), where 32.1% of the patients in the intervention group became adherent compared to 9.4% in the control group. By using the generalized linear model–Ordinal logistic regression, the adherence scores were found to be decreased (better adherence) upon the implementation of the MMR service among the intervention group. In other words, adherence scores were improved significantly by the factor 0.134, P < .001 in the intervention group upon MMR service implementation. However, this improvement was not significant among the control group (P = 0.716, generalized linear model–Ordinal logistic regression). Moreover, a significant correlation was found between the improvement in the adherence score and the implementation of the MMR service among the intervention group (r = 0.506, P < .001), but not the control group (r = 0.035, P = 0.720).

There was no significant difference with regards to patients' commitment to pharmacist advice between the two groups at baseline; at follow-up, a significant difference was found (P = 0.01, Chi-square test). Around 73% of intervention group patients were committed to the pharmacist advice at baseline compared to 88% commitment at follow-up.

Causes of non-adherence to medications were assessed at baseline visits. More than one third of the patients in the intervention group and the control group reported that the price was the main cause for non-adherence to their medication (39.7% and 37.7% respectively). Other causes for patients' non-adherence are listed in Table 2. There was no statically significant difference between the two groups regarding the reported causes for non-adherence to medications at baseline (P = 0.764, Chi-square test).

Regarding the degree of patients' commitment to pharmacist advice, there was no significant difference between the two groups at baseline. At follow-up, a significant improvement

**Table 2. Main causes for non-adherence to medications at baseline.**

| Causes of non-adherence, n(%) | Intervention group (n = 53) | Control group (n = 53) |
|---|---|---|
| Price | 21 (39.7) | 20 (37.7) |
| Timing of dose | 1 (1.9) | 2 (3.9) |
| Forgetting | 7 (13.2) | 8 (15.1) |
| I do not like medicines | 4 (7.5) | 8 (15.1) |
| Medicine does not work | 4 (7.5) | 5 (9.4) |
| Side effects | 9 (17) | 5 (9.4) |
| High number of pills every day | 7 (13.2) | 5 (9.4) |

n: Number of patients

was found in the intervention group but not the control group. (P = 0.01, Chi-square test) (Table 3).

## Knowledge of medications at baseline and follow-up

At baseline, no significant difference was found between the intervention and the control groups with regards to 'Knowledge of medications mean scores. Across the study, there was a significant difference within the intervention group, but not the control group (Fig 3). From the factors which were assessed to investigate their effect on the knowledge score change (number of medications, number of medical conditions, age, and education level), none of these factors was found to influence the score change among patients.

Upon comparison between the intervention and the control groups regarding the five domains (using Chi-square test) included in the ad hoc knowledge about drug therapy questionnaire [29], there was no significant difference between the two groups in the five domains at baseline. However, significant differences resulted at follow-up regarding all domains (Table 4). By using the generalized linear model–Ordinal logistic regression, the knowledge about drug therapy scores were found to be decreased (better knowledge) upon the implementation of the MMR service among the intervention group. In other words, knowledge about medications scores were improved significantly by the factor 0.054, P < .001 in the intervention group upon MMR service implementation. However, this improvement was not

**Table 3. Commitment to pharmacist advice at baseline and follow-up.**

| Parameter | Intervention group n = 53 | | Control group n = 53 | | P-value |
|---|---|---|---|---|---|
| Commitment to pharmacist advice at baseline, n (%) | Never | 0 (0.0) | Never | 1 (1.9) | 0.495[1] |
| | Rare | 4 (7.5) | Rare | 7 (13.2) | |
| | Sometimes | 10 (18.9) | Sometimes | 10 (18.9) | |
| | Usually | 17 (32.1) | Usually | 20 (37.7) | |
| | Always | 22 (41.5) | Always | 15 (28.3) | |
| Commitment to pharmacist advice at follow-up, n (%) | Never | 0 (0.0) | Never | 1 (1.9) | 0.010[1] |
| | Rare | 0 (0.0) | Rare | 6 (11.3) | |
| | Sometimes | 6 (11.3) | Sometimes | 13 (24.5) | |
| | Usually | 21 (39.6) | Usually | 19 (35.8) | |
| | Always | 26 (49.1) | Always | 14 (26.5) | |

n: Number of patients

[1] Analysis by Chi-square test

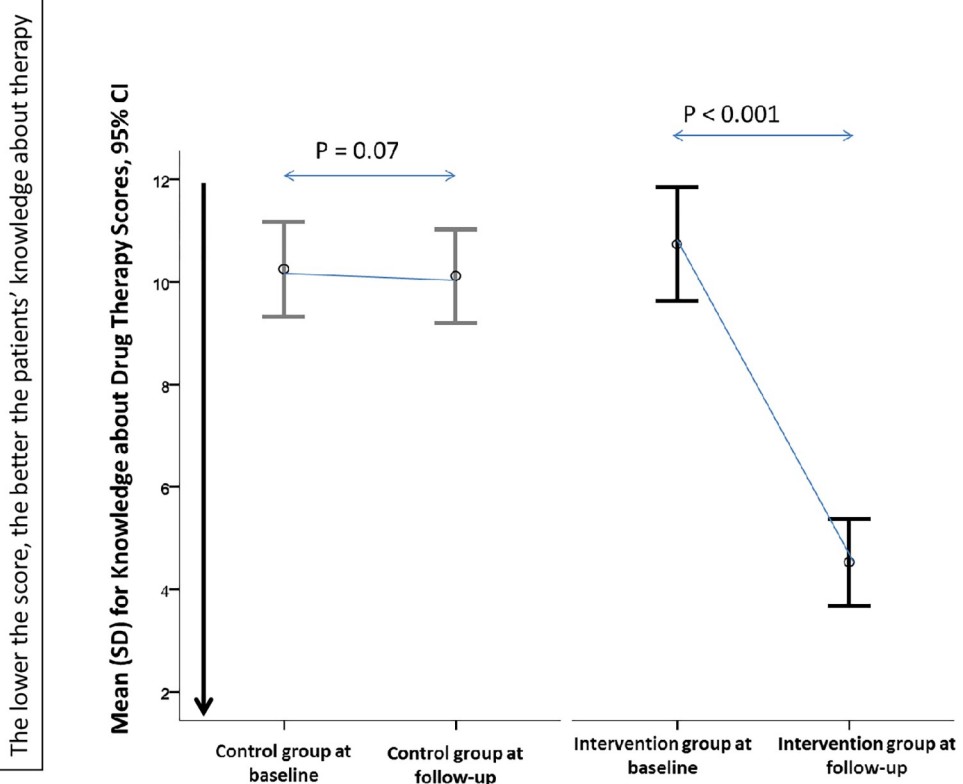

*P values by paired sample t-test

**Fig 3. The knowledge about drug therapy mean changes at baseline and follow-up.**

significant among the control group (P = 0.769, generalized linear model–Ordinal logistic regression). Moreover, a good correlation was found between the improvement in the knowledge scores and the implementation of the MMR service among the intervention group (r = 0.671, P < .001), but not the control group (r = 0.028, P = 0.772).

## Discussion

The current study showed that the provision of the pharmacist-lead MMR service can lead to significant improvements in the adherence to medications and knowledge of medications for Syrian refugees living in Jordan. To our knowledge, this randomized controlled study is the first to be conducted for Syrian refugees around the world, assessing the impact of the MMR service on their adherence and knowledge of their chronic medications, decreasing the number of TRPs they suffer from [28]. The study emphasizes the important role that clinical pharmacists can play in such crisis to help refugees to have better management of their health conditions and hence lives.

The Syrian conflict has been declared as a humanitarian crisis since 2011, leading to a health disaster for the country and for the region alike [32]. Overall, refugees are the weakest and most vulnerable group in the conflict setting, and therefore, their medical needs are expected to require urgent assistance [33]. Many studies conducted during the last years reported that pharmaceutical care interventions lead in most cases to desirable clinical and financial outcomes [34]. Pharmacists were found to help in improving patients' health by reducing

**Table 4. Comparison of the 'knowledge about drug therapy' five domains between the two groups.**

| Parameter n (%) | Intervention group n = 53 | | Control group n = 53 | | P-value | P-value |
|---|---|---|---|---|---|---|
| | Baseline | Follow-up | Baseline | Follow-up | Baseline | Follow-up |
| **Know the scientific name of medication** | | | | | 0.6[1] | < 0.001[1] |
| • I know very well | 0 (0.0) | 5 (9.4) | 0 (0.0) | 0 (0.0) | | |
| • I know to some extent | 2 (3.8) | 14 (26.4) | 5 (9.4) | 5 (9.4) | | |
| • I know the name for some of my medications only | 10 (18.9) | 15 (28.3) | 7 (13.2) | 8 (15.1) | | |
| • I don't know | 15 (28.3) | 11 (20.8) | 16 (30.2) | 16 (30.2) | | |
| • I don't know at all | 26 (49.1) | 8 (15.1) | 25 (47.2) | 24 (45.3) | | |
| **Know the generic name of medication** | | | | | 0.961[1] | < 0.001[1] |
| • I know very well | 0 (0.0) | 12 (22.6) | 0 (0.0) | 0 (0.0) | | |
| • I know to some extent | 9 (17) | 19 (35.8) | 9 (17) | 9 (17) | | |
| • I know the names for some of my medications only | 17 (32.1) | 16 (30.2) | 17 (32.1) | 18 (34) | | |
| • I don't know | 14 (26.4) | 6 (11.3) | 12 (22.6) | 13 (24.5) | | |
| • I don't know at all | 13 (24.5) | 0 (0.0) | 15 (28.3) | 13 (24.5) | | |
| **Know how to take the medication** | | | | | 0.092[1] | < 0.001[1] |
| • I know very well | 9 (17) | 37 (69.8) | 9 (17) | 9 (17) | | |
| • I know to some extent | 23 (43.4) | 11 (20.8) | 21 (39.6) | 21 (39.6) | | |
| • I know how to take some of my medications only | 14 (26.4) | 5 (9.4) | 16 (30.2) | 16 (30.2) | | |
| • I don't know | 2 (3.8) | 0 (0.0) | 7 (13.2) | 7 (13.2) | | |
| • I don't know at all | 5 (9.4) | 0 (0.0) | 0 (0.0) | 0 (0.0) | | |
| **Know when to take the medication** | | | | | 0.446[1] | < 0.001[1] |
| • I know very well | 7 (13.2) | 41 (77.4) | 8 (15.1) | 8 (15.1) | | |
| • I know to some extent | 24 (45.3) | 10 (18.9) | 23 (43.4) | 24 (45.3) | | |
| • I know when to take some of my medications only | 13 (24.5) | 2 (3.8) | 18 (34) | 17 (32.1) | | |
| • I don't know | 7 (13.2) | 0 (0.0) | 4 (7.5) | 4 (7.5) | | |
| • I don't know at all | 2 (3.8) | 0 (0.0) | 0 (0.0) | 0 (0.0) | | |
| **Know why taking the medication** | | | | | 0.398[1] | < 0.001[1] |
| • I know very well | 2 (3.8) | 32 (60.4) | 4 (7.5) | 4 (7.5) | | |
| • I know to some extent | 11 (20.8) | 15 (28.3) | 12 (22.6) | 11 (20.8) | | |
| • I know why I take some of my medications only | 29 (54.7) | 6 (11.3) | 31 (58.5) | 34 (64.2) | | |
| • I don't know | 8 (15.1) | 0 (0.0) | 6 (11.3) | 4 (7.5) | | |
| • I don't know at all | 3 (5.7) | 0 (0.0) | 0 (0.0) | 0 (0.0) | | |

**n:** Number of patients

**1** Analysis by Chi-square test

medication-related side effects and encouraging medication adherence, decreasing physician visits, hospital admissions, and amending the whole primary care delivery [35]. But this study comes to show that the same can happen in a population of great need of this help, the Syrian refugees.

Adherence to medications is a critical part of general patient care. It is substantial for reaching the targeted goals for any patient, yet alone the refugees [36]. By opposition, non-adherence to medications is considered a main challenge to healthcare, with approximately 50% of patients reported not taking their chronic medications as prescribed 12 months after starting their therapy [37, 38]. The likely negative impact of non-adherence to medications on morbidity and mortality is well documented [39]. In the current study, adherence to medications was assessed utilizing a previously developed, validated and published questionnaire by AbuRuz et al. [29], self-reported by patients, allowing feasibility and ease of use by patients with

different diseases, which is vital for the success of this assessment [29]. The changes in the non-adherence mean scores in the intervention group was significant, indicating the positive impact of the MMR service on participants' adherence to their medications.

It is known from the literature that pharmacist interventions across the years, in its various forms, can improve participants' adherence., A prospective randomized study was found to significantly improve adherence of participants in the intervention group receiving a pharmacist counselling session compared to control group (92.1% versus 23.7%) [40]. A Jordanian study showed that the pharmaceutical intervention resulted in significantly lower proportion of non-adherent diabetic participants in the intervention group [41]. Moreover, two studies were conducted in 2016 and revealed that the MMR and the comprehensive medication management (CMM) services -respectively- resulted in significant improvement in medication adherence among the studies' samples [19, 39]. Such findings support the results of the current study, where adherence to medications was improved significantly as a result of pharmaceutical care services such as the MMR service. The adherence percentages among both groups at base line were really low, and this can be explained-in part- by the nature of the study sample and the harsh life conditions that they live in, which lack the fundamental elements of normal life. Although the percentage of adherent patients form the intervention group has been improved to reach 32%, yet this percentage is not sufficient for proper health management. However, further follow up visits/calls, which include educational and behavioural intervention (such as suggesting more appropriate dosing times for patients, educate them how to arrange the medications in pill-organizer boxes for easier use, educate them how set reminders for their medications different times) for the refugees could affect positively the adherent percentages and could result in more improvements [42]. In addition, the study was conducted over three months only; it is suspected that with longer study duration more improvement in medication adherence would be resulted.

Knowledge of medications is essential for proper adherence and good management of chronic conditions to be reached. Participants' knowledge of their medications showed a significant improvement across the study for the intervention group, but not the control group. A previous study showed that a significant improvement in the knowledge of medications was seen for participants in the intervention group as a result of a pharmacist-led intervention [43]. In addition, a large number of American studies (n = 298) included in a meta-analysis review proved that 57.1% of these studies favour the positive impact of pharmacists on the participants' knowledge of their medications [44].

The approach described here, i.e. wherein the MMR service is delivered to a population (the refugees) who needs it and can benefit from it the most, represents a virtuous strategy to target and maximize the impact of health care professional services. The extension of this model to large scale would sharply address and support some of the targets of the Sustainable Development Goal (SDG) 3 "*Ensure healthy lives and promote well-being for all at all ages*". In particular, delivery of MMR services to refuges would contribute to achieve the SGD targets 3.4 "*By 2030, reduce by one third premature mortality from non-communicable diseases through prevention and treatment and promote mental health and well-being*" and 3.8 "*Achieve universal health coverage, including financial risk protection, access to quality essential health-care services and access to safe, effective, quality and affordable essential medicines and vaccines for all*". This comes in line with the WHO recommendations [45] supporting the execution of the recently published report by the High-Level Commission on Health Employment and Economic Growth [46] calling for "ambitious solutions to ensure that the world has the right number of jobs for health workers with the right skills and in the right places to deliver universal health coverage". A focus on trained universal health-workforce that has the ability to deliver different healthcare services in humanitarian settings was highlighted [46].

From the study results, a remarkable improvement for the overall TRPs [28], adherence, and knowledge of medications were shown among the Syrian refugees. In addition, physicians and patients who were involved in this study were highly satisfied about this service [28]. Consequently, the implementation of such service (MMR) in the clinical settings in Jordan and delivering this service routinely for this population would be well accepted and would enhance the overall health situation for refugees and hopefully for other populations.

## Study limitations

The pre-specified sample size for the study was not achieved due to time constraints, to the difficulty that researcher faced in reaching refugees' houses to complete the home visits at baseline and follow-up, and to the entire challenging nature of the study (deplorable living conditions which were seen throughout the whole data collection process). It is plausible that this reduction in sample size and the subsequent increase in the margin of errors can affect the confidence level and thus the results obtained from the study. A longer follow up could have added new insights to the results. Regarding the subjective nature of the study's outcome measures, some limitations can be associated to their use such as possible related biases, difficulty of results' interpretation and poor correlation to some independent factors. However, these subjective outcome measures are being increasingly used in many studies and the usefulness of using them were approved in the literature. We did not assess health literacy abound the current medications used by patients which could have further explained improvements in medication knowledge improvements. We did not study recommendations rejected by the physicians nor the ones not applied by the patients. This would be important for future studies to assess the value of physician-patient involvement in decision making [43]. Moreover, it is possible that patients were subjected to social acquiesce bias, which could influence their responses to the questionnaire at follow-up. However, significant improvements were not only reported between baseline and follow-up assessments, but also between the intervention and the control group at follow-up. In addition, the fact that the recruited patients originally chose to participate in this study, and consequently they were willing to accept and consider the suggested changes which were made by the physicians, this could affect the outcomes of the study and could create some type of bias.

Although the research group have tried to limit the confounding factors in this study, yet the different time spent with each patient at home visit, and the different time required by the clinical pharmacist for each patient's counselling and education could have effect on the outcomes.

## Conclusion

This study revealed that many refugees with numerous medications and chronic health condition have numerous TRPs [28], low adherence and low knowledge of their medications.

Despite that the recruitment was for 85% of the calculated sample size, the findings do nevertheless suggest that the significant improvements in adherence and knowledge to treatment was reached as a result of the MMR service provided by the clinical pharmacist. Pharmaceutical care is an essential aspect of patient care and communication between healthcare providers can help enhance this care with expressively upgraded consequences for the individual refugee patient. While this understanding is intuitively accurate, few studies have apprehended the findings to report these benefits. Therefore, this study provides an important contribution to this understanding of medication management for displaced refugee communities. The positive outcomes of this study emphasizes the important role pharmacists can play in conveying a vital and required service in a vast humanitarian setting globally.

## Supporting information

**S1 Checklist. CONSORT 2010 checklist of information to include when reporting a randomised trial**∗**.**
(DOC)

**S1 File. IRB ethical approval.**
(DOCX)

**S2 File. Original study protocol.**
(DOCX)

## Author Contributions

**Conceptualization:** Majdoleen Alalawneh, Alberto Berardi, Nabeel Nuaimi.

**Data curation:** Majdoleen Alalawneh, Iman A. Basheti.

**Formal analysis:** Majdoleen Alalawneh, Iman A. Basheti.

**Funding acquisition:** Majdoleen Alalawneh.

**Methodology:** Majdoleen Alalawneh, Iman A. Basheti.

**Project administration:** Majdoleen Alalawneh.

**Software:** Majdoleen Alalawneh.

**Supervision:** Iman A. Basheti.

**Writing – original draft:** Majdoleen Alalawneh.

**Writing – review & editing:** Alberto Berardi, Nabeel Nuaimi.

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
