## [Decision Letter · Decision Letter 0]

16 Mar 2021

PONE-D-20-26688

Improving Syrian refugees’ knowledge of medications and adherence following a randomized control trial assessing the effect of a medication management review service

PLOS ONE

Dear Dr. Basheti,

Thank you for submitting your manuscript to PLOS ONE. After careful consideration, we feel that it has merit but does not fully meet PLOS ONE’s publication criteria as it currently stands. Therefore, we invite you to submit a revised version of the manuscript that addresses the points raised during the review process.

The manuscript has been evaluated by four reviewers, and their comments are available below.

The reviewers have raised a number of concerns that need attention, and they request additional information on methodological aspects of the study and revisions to the statistical analyses.

Could you please revise the manuscript to carefully address the concerns raised?

We look forward to receiving your revised manuscript.

Kind regards,

Vanessa Carels

Staff Editor

PLOS ONE

Journal Requirements:

2. Please include the CONSORT checklist as a Supporting Information file.

3. In the Methods and study limitations sections, please discuss the discrepancy between the sample size in the protocol and manuscript.

4. We noticed you have some minor occurrence of overlapping text with the following previous publication(s), which needs to be addressed:

- https://pubmed.ncbi.nlm.nih.gov/32256897/

he text that needs to be addressed involves the Abstract, the first two paragraphs of the Discussion, and the Conclusion.

In your revision ensure you cite all your sources (including your own works), and quote or rephrase any duplicated text outside the methods section. Further consideration is dependent on these concerns being addressed.

Reviewers' comments:

Reviewer's Responses to Questions

**Comments to the Author**

1. Is the manuscript technically sound, and do the data support the conclusions?

Reviewer #1: Yes

Reviewer #2: No

Reviewer #3: Partly

Reviewer #4: No

2. Has the statistical analysis been performed appropriately and rigorously? 

Reviewer #1: Yes

Reviewer #2: No

Reviewer #3: Yes

Reviewer #4: No

3. Have the authors made all data underlying the findings in their manuscript fully available?

Reviewer #1: Yes

Reviewer #2: No

Reviewer #3: Yes

Reviewer #4: No

4. Is the manuscript presented in an intelligible fashion and written in standard English?

Reviewer #1: Yes

Reviewer #2: Yes

Reviewer #3: Yes

Reviewer #4: Yes

5. Review Comments to the Author

Reviewer #1: Thank you for giving me the opportunity to review this nice paper, which is also important for the pharmacy field.

I would like to congratulate the authors for their efforts and for this nice work. Kindly find my comments attached inside the PDF file. Good luck.

Reviewer #2: An interesting RCT on a very valuable topic and population. There are a few methodological concerns and limitations not discussed. Major omission is the lack of TRP results discussed in the manuscript, despite them being prominent in the abstract. Please see attachment for further comments.

Reviewer #3: Authors´ comments

Thank you for allowing me to review such an interesting work. Here are my comments on the manuscript:

ABSTRACT

Line 44: The word “chroesacnic” should be modified (chronic?)

Line 47: The objectives of the paper are “to evaluate the effect of the MMR service on adherence and on knowledge”. The identification of TRPs is not an objective of the paper. However, along the abstract, TRPs are described in methods, results (they are measured at baseline and at the end of the three months period) and cited as a conclusion. Conversely, in the body of the manuscript TRPs are hardly cited. In my opinion, authors should clarify the role of TRPs in this piece of research.

Line 55: “Mediation” should be changed into “medication”

Line 63: TRPs are also included as a conclusion. So, in my opinion, they should be one of the objectives of the manuscript.

INTRODUCTION

Very interesting description of the situation of Syrian refugees and the great amount of health problems they have to deal with, among other type of problems. However, for the purpose of this paper, I lack a further explanation on how these refugees manage to live there: do they live as a group in refugees camps, or do they locate or buy particular homes? Do they manage to find and have a job in Jordan or they live from charity? As in the paper the researcher goes to “homes”, in my opinion and taking into account I don’t have a proper knowledge of the real situation of this collective, this issue (socioeconomic situation of the refugees) should be clarified.

The other issue with the introduction is that there is no reference to TRPs. Although they’re not included as an objective (I think perhaps they should) TRPs play a role in the paper, as they are included both in methods, and in conclusions.

OBJECTIVES

From the above I suggest to include TRPs evaluation as an objective. (In fact, in the ClinicalTrials web it is said that one of the objectives of the research is to evaluate the impact of MMR service)

Line 129. It´s not clear for me the sentence: “The registration of this clinical study delayed, as it is not requested by Jordan). In my opinion, it should be clarified.

Line 138: It is no clear for me which is the “original study protocol” to which the authors refer to. The “original study protocol” should be clearly stated or described and supported by a bibliographic reference.

Line 132-142: I wonder how the participants were approached and selected? There was any systematic design to identify possible participants or just those who went to the clinics in a particular period of time were contacted?

Line 183: Adherence: The authors say that “adherence to medication was assessed by a developed and published questionnaire” (Reference #21). However, the questionnaire used by AbuRuz et al (21) is the well-known 8 items Morisky questionnaire. AbuRuz et al described the questionnaire they used as “a validated translation of the scale developed by Morisky et al”. In my opinion this should be clearly stated and referenced.

Line 198: Knowledge. Again, the authors refer to the paper by AbuRuz (21) while explaining the methodology related to the measurement of knowledge about chronic medication. Surprisingly, they say that they used a “validated and published questionnaire” but in the cited paper (21) no questionnaire is used. In fact, measuring knowledge wasn´t an objective of the AbuRuz´ paper. In line 267, the authors make a reference to the “five domains of the knowledge about drug therapy´ questionnaire” which is shown in table 4. That affirmation should be supported by a literature reference different from AbuRuz´(21). In addition, if a “validated questionnaire had been used” any further modification should be well explained and tested (at least piloted) in order to not to get their properties loosen. I suggest the authors should rephrase this section explaining the procedure and their bibliographic support.

DISCUSSION

Line 276: Although TRPs are not included as an objective and they´re not described in the results section, the authors states that: “…decreasing the number of TRPs they suffer from”. This assertion should come from previously described results. So, it should be rephrased.

From line 300 to line 314 there is a comparison of the adherence results with those of four other papers (15, 30, 31, 32, 33), comparison that in my opinion makes the discussion too long. I recommend shortening this paragraph.

On the other hand, the authors found that at baseline, adherence ranged between 13.2% in the IG and the 11.3% in CG (p value = 0.540). After the follow up, adherence improved until the 32.1%. These figures mean that at baseline non adherence rates among Syrian refugees were higher than 80% (IG: 86.8%; CG:88.7%) and that after the intervention a 67.9% of non-adherence was achieved in the IG. I think these figures should be commented in the Discussion because they show really high rates of non-adherence at baseline (and even at the end of the study, being it very successful).

Another issue that, in my opinion, deserves a comment in the discussion is that only a 14.5% of the non-adherent patients (n=15; table 2) were really unintentional.

CONCLUSION

Line 347: Again, there is a reference to TRPs which haven´t been described in the results section.

BIBLIOGRAPHY

Literature references are repeated and have different numbers. References should be revised.

Line 363 – 463 References 1 – 36

Line 555 – 658 References 1 – 37

Line 659 – 760 References 1 – 37

Line 761 – 864 References 1 – 37

Line 865 – 968 References 1– 37

Line 969 – 1057 References 1 – 34

Reviewer #4: PONE-D-20-26688: statistical review

SUMMARY. This study evaluates the effect of medication management review (MMR) on adherence to treatment

therapy and knowledge about chronic medications for Syrian refugees residing in Jordan. The sample was randomly split in two groups (control and MMR-treated) and the response variables (adherence and knowledge) were measured in two occasions by two questionnaires. While this randomization case-control study seems well designed, I am seriously concerned about the way the response variables are defined (major issue 1), interpreted (major issue 2) and modelled (major issue 3). Unfortunately, these issues require a complete revision of the statistical analysis, which can be however mostly performed by using SPSS, the software exploited by the authors. I also list a couple of specific points that should be addressed.

MAJOR ISSUES

1. Qualitative answers are transformed into numeric values. Mapping the support of an ordered factor to the support of a numeric variable is in general not recommended, as it relies on a arbitrary choice. It is also not necessary, because generalized linear models (GLM) for ordered variables are available for analysis (also in SPSS, which is the software used in this paper).

2. The response variable are obtained by summing scores derived from different questions. Summing heterogeneous scores that reflect different dimensions of adherence and knowledge make the interpretation of the response variable rather difficult: for instance, two subjects with different patterns of answers could obtain the same score. This is tolerable when the items are exchangeable. Is it the case here? If it is not, then the response variable should be treated as a multivariate response. An item response model is the state-of-the art method to analyze multiple items in a questionnaire.

I don't know whether SPSS includes these models, but they are available in many software packages such as R and SAS.

3. Variable adherence is not a continuous variable: it is a discrete variable with range 0-32. Perhaps (not clear from text, see specific point 2) variable knowledge is discrete too. The t-test is therefore not fully appropriate in this context. Generalized linear mixed models (available in SPSS, which is the software used in the paper) provide the state-of-art strategy to compare discrete responses in a longitudinal setting.

SPECIFIC POINTS

1. lines 183 - 189. Only 5 questions are described, while the questionnaire should include 8 questions: please clarify.

2. Lines 204-205. The continuous scale out of 5 is not described. What are the possible scores? The integers 1 to 5? Or any real number between 1 and 5? Please clarify.

6. PLOS authors have the option to publish the peer review history of their article (what does this mean?). If published, this will include your full peer review and any attached files.

Reviewer #1: **Yes: **Souheil Hallit

Reviewer #2: No

Reviewer #3: **Yes: **miguel angel gastelurrutia or Gastelurrutia MA

Reviewer #4: No

---

## [Author Response · Author response to Decision Letter 0]

2 Jul 2021

Reviewer 1

Comment Response

‘’Why have the authors waited so long to write the article? It's been almost 5 years’’. Thank you for the question.

This RCT included huge data/results about the Treatment Related Problems (TRPs), quality of life, anxiety, adherence, and knowledge about medications for the Syrian refugees. These pieces of data/work were published consecutively from 2019 onwards. Writing manuscripts and following the publication process need time from authors who were sometimes busy with other commitments.

Please note that no changes were done on the manuscript in response to this comment. 

In the method section, under the Adherence to Medication subtitle, it was commented:

‘’what's the name of it?’’

 This questionnaire is called ''eight-item Morisky Medication Adherence Scale (MMAS-8)''. The name of the questionnaire and additional reference was added to the sentence as follow (please see the modifiable Word version of the manuscript):

‘’eight-item Morisky Medication Adherence Scale (MMAS-8)[References] ‘’

In the method section, under the Knowledge about Chronic Medication subtitle, it was commented:

‘’name of the scale?’’

 It is name is “Knowledge about Drug Therapy” questionnaire, which was validated by AbuRuz et al (2011). The name was added to the modifiable Word version of the manuscript, as follow:

 “Knowledge about Drug Therapy”

In the statistical analysis section, it was commented:

‘’why not the McNemar test?’’

 The decision of using specific test for specific data type was done by consensus of the research team at that time. I think that both tests (Chi-square and McNemar) can be used for our data (data type/purpose). Chi-square test was used here as it compared a categorical data between two groups. 

Please note that no changes were done on the manuscript in response to this comment.

In the Sample Size section, it was commented:

‘’I think this paragraph should be placed after the study design.’’

 The paragraph is moved after the Study design, as suggested (please see the modifiable Word version of the manuscript). 

In the Sample Size section, it was commented:

‘’the minimal sample size needed was 138 as calculated. I see that the number you recruited is lower than that. Does it mean that your study is underpowered?’’

 Thank you for the question. 

Initially and due to the time restriction, 123 patients were visited at their homes during the enrolment period (to check their eligibility for participation), and 116 of them were recruited in the study as they met the inclusion criteria. Although the needed sample size has not been reached, the results were auspicious with the participants. However, this could affect the power of the study, as you said.

The following sentences have been added to the limitation section in response to this comment:

‘’The pre-specified sample size for the study was not achieved due to time constraints. 

This reduction in sample size and the subsequent increase in the margin of errors can affect the confidence level of the study.’’

In the Results section:

 ‘’Was’’ The word ‘’were’’ is replaced by ‘’was’’ in the modifiable Word version.

In the Results section, it was commented:

‘’Can the authors assess the factors associated with change in adherence and knowledge at T1 vs T0? (through repeated measures ANOVA).

I think this would enrich the results and discussion more.’’

 The following paragraph is added to the manuscript (at adherence result section):

‘’From the factors which were assessed (number of medications, number of medical conditions, age, and education level), the number of medications was the only factor which affected the change in adherence scores between baseline and follow-up among patients (P = 0.009)’’

Also, the following paragraph is added to the manuscript (at knowledge result section):

‘’From the factors which were assessed to investigate their effect on the knowledge score change (number of medications, number of medical conditions, age, and education level), none of these factors was found to influence the score change among patients’’

At the end of the discussion section, it was commented:

‘’Maybe authors can add a paragraph about the clinical implications of this study.’’

 The following paragraph is added to the manuscript (at the end of Discussion section):

‘’From the study results, a remarkable improvement for the overall TRPs [Reference], adherence, and knowledge about medications were shown among the Syrian refugees. In addition, physicians and patients who were involved in this study were highly satisfied about this service [Reference]. Consequently, the implementation of such service (MMR) in the clinical settings in Jordan and delivering this service routinely for this population would be well accepted and would enhance the overall health situation for refugees and hopefully for other populations.’’

In the Study Limitations section, it was commented:

‘’ There are other biases to discuss in this section: Information bias, Social desirability bias, Residual confounding bias’’

 The suggested biases are discussed. The following paragraph was added to the manuscript, as follow:

‘’Moreover, it is possible that patients were subjected to social acquiesce bias, which could influence their responses to the questionnaire at follow-up. However, significant improvements were not only reported between baseline and follow-up assessments, but also between the intervention and the control group at follow-up. Although the research group have tried to limit the confounding factors in this study, yet the different time spent with each patient at home visit, and the different time required by the clinical pharmacist for each patient’s counselling and education could have effect on the outcomes. ‘’

In the References section, it was commented:

‘’The references are repeated multiple times. Please adjust accordingly.’’

 The repeated references are deleted, as suggested.

Reviewer 2

Comment Response

Line 3: Running title – remove capital R on refugees’ Corrected as suggested

Line 44: what is chroexacnic? Should it be chronic

 It is changed to ‘’chronic’’

Line 48: I think knowledge of chronic medications may sound better than knowledge about chronic medications ‘’knowledge about chronic medications’’ has been changed to ‘’ knowledge of chronic medications’’ in the indicated line and through the manuscript.

Line 123: “following delivery of the service” may sound better Changed as suggested

Line 129: I’m not sure what the point of registering the study four years after it was conducted. Can this be made clearer and not just say “delayed” This RCT has not been registered due to time restriction issue at the time of conduction, and due to the fact that the registration of the clinical trials is not mandatory in Jordan. However, the registration for this RCT was asked for by PLOS ONE journal to proceed with the submission. Consequently, the authors proceeded with the registration process, and the submission process after that.

The following word are added to the manuscript:

‘’ and due to time constraints’’

Line 139: What impact does the reduced sample size have on confidence of findings? Please discuss in limitations section. The following sentences have been added to the limitation section:

‘’The pre-specified sample size for the study was not achieved due to time constraints. 

This reduction in sample size and the subsequent increase in the margin of errors can affect the confidence level of the study.’’

Line 151: Can you provide further detail about the clinical pharmacist. Was there just one for the study? How many years of experience do they have? Do they have additional qualifications?

 The following sentence is added to the Method section (under the Study design and clinical setting subheading):

‘’One clinical pharmacist, who has four years’ experience in the field (with no additional qualifications), has recruited the Syrian refugees and has delivered the MMR service to them.’’

Line 175: were recollected (not where) Corrected

Line 179: Was there any thought into the appropriateness of the data collection questionnaires for the refugees? I.e. did they have adequate health literacy to use these questionnaires? The following sentence is added to the manuscript (under Data collection tool subheading):

‘’Patients were asked to answer these questionnaires by themselves. However, in case of a certain patient was unable to read/understand the given questionnaire, the appropriate help was provided by the clinical pharmacist.’’

Line 188: the scale appears to represent an ordinal scale, not a continuous scale. Can you please comment on the appropriateness of treating as a continuous scale? Thank you for the comment. 

adherence and knowledge of medication are analysed as a Likert scale, instead of treating them as continuous scales. 

Consequently, the generalized linear model (SPSS) is used to analyse the ordered variables in both questionnaires, as suggested. Changes in Method and Results sections were done accordingly.

Line 191: was the main cause of non-adherence an open question or did patients choose from a set list (as it seems from results in Table 2) – please clarify

 The following sentence was added to the manuscript for clarification (under Adherence to medication subheading):

‘’The possible listed reasons for non-adherence were as follow: price, timing of dose, forgetting, I do not like medicines, medicine does not work, side effects, high number of pills every day.’’

Line 202: Was medication knowledge assessed per medication? Or as an aggregate measure? Also, was the patient-reported level of medication knowledge checked by the pharmacist? Medication knowledge was assessed as an aggregate measure. 

The questionnaires were answered by patients themselves, with no checking by the clinical pharmacist.

The following sentence was added:

‘’Medication knowledge was assessed as an aggregate measure for all medications per patient.’’

Line 209: Were all continuous variables expressed as mean SD, was consideration given to parametric vs non-parametric distribution? The adherence questionnaires are typically skewed towards adherence. The data in the study was normally distributed (the normality of distribution was assessed using Kolmogrov-Smirnov test), so the parametric tests were used to analyze the variables in the study.

For the adherence scale, a certain patients was considered either adherent or non-adherent, depending on him/his questionnaire scores. The patients were considered under-adherent if they scored 1.0 or more in the total in the questionnaire (Morisky et al. 1986). This clarification is added to the methodology section with the corresponding reference (under adherence to medication subheading) as follow:

‘’The patients were considered under-adherent if they scored 1.0 or more in the total [Ref]’’

Moreover, the following sentence is added to the Statistical analysis section for further clarification:

‘’Likert scale scores (for adherence and knowledge of medication questionnaire) were analysed using the generalized linear model.’’

Line 226: states 10 participants refused to continue however the numbers and figure 1 suggest this number was 7 participants. Please clarify. The sentence is not accurate, and it is corrected according to Fig 1.

7 patients refused to participate, and 3 patients were dropouts (please see the modified version of the manuscript).

Line 223: Was the mean number of medications and chronic conditions similar across groups. Can this be stated either in text or in table 1. The number of medications and the number of chronic conditions were similar in both groups. The following sentence is added to the manuscript to clarify this (in the Results section):

‘’There were no significant differences between the two groups’ participants with regards to the number of medications (P = 0.099) and the number of chronic conditions (P = 0.054).’’

Line 238: Please keep consistent terminology throughout manuscript – active group or intervention group. The word ‘’active’’ is changed to ‘’intervention’’

Results general: were there any patterns in the people who improved, e.g. based on baseline demographics. Was the service more effective for those that took more medications, had more complex regimens, had better education etc?

 The following paragraph is added to the manuscript (at adherence result section):

‘’From the factors which were assessed (number of medications, number of medical conditions, age, and education level), the number of medications was the only factor which affected the change in adherence scores between baseline and follow-up among patients (P = 0.009)’’

Also, the following paragraph is added to the manuscript (at knowledge result section):

‘’From the factors which were assessed to investigate their effect on the knowledge score change (number of medications, number of medical conditions, age, and education level), none of these factors was found to influence the score change among patients’’

TRPs not mentioned in the results at all – yet they are listed in the protocol and abstract. Have they been omitted intentionally or accidently? Please add the results into the manuscript. The TRPs’ results are published previously. The following sentence with the corresponding reference is added into the manuscript to clarify this (Results section):

‘’The outcomes regarding the impact of the MMR on the TRPs’ type and frequency are reported elsewhere [reference]’’

Line 278: improve lives is not supported by this study as quality of life was not reported. I note that quality of life is in your protocol – are those results going to be published separately? Please clarify. The quality-of-life results are published previously. Please see the following reference:

Alawneh MA, Nuaimi N, Abu-Gharbieh E, Basheti IA. A randomized control trial assessing the effect of a pharmaceutical care service on Syrian refugees’ quality of life and anxiety. Pharmacy Practice. 2020;18(1) 

Please note that no changes were done on the manuscript in response to this comment.

Line 339: Further limitations to discuss include the subjective nature of your outcome measures, the fact that those that chose to participate may have been more receptive to advice/change, not meeting sample size etc The following paragraphs are added to the Limitation section:

‘’Regarding the subjective nature of the study’s outcome measures, some limitations can be associated to their use such as possible related biases, difficulty of results’ interpretation and poor correlation to some independent factors. However, these subjective outcome measures are being increasingly used in many studies and the usefulness of using them were approved in the literature.’’

‘’ In addition, the fact that the recruited patients originally chose to participate in this study, and consequently they were willing to accept and consider the suggested changes which were made by the physicians, this could affect the outcomes of the study and could create some type of bias.’’

‘’The pre-specified sample size for the study was not achieved due to time constraints. 

This reduction in sample size and the subsequent increase in the margin of errors can affect the confidence level of the study.’’

Line 342: why were recommendations approved/rejected not reported? What is the approval rate (an outcome specified in protocol)? The approved/rejected recommendations were not studied, unfortunately. However, this is mentioned in the limitation section of this study. 

The approval rate of pharmacist’s recommendations by the contacted physicians was high (82.9 %). This outcome is published previously. The following sentence with corresponding reference is added to Results section to clarify this:

‘’ The approval rate of pharmacist’s recommendations by the contacted physicians was high (82.9 %) [Ref].’’

Discussion: consider discussing next steps, i.e. how could the intervention be further improved. 32% adherent is still very low and definitely not sufficient to reach adequate management of health. Consider comparing your intervention to others:

https://www.cochranelibrary.com/cdsr/doi/10.

1..2/14651858.CD012419.pub2/full

 The following paragraph is added to the Discussion section:

‘’Although the percentage of adherent patients form the intervention group has improved to reach 32%, yet this percentage is not sufficient for proper health management. However, further follow up visits/calls, which include educational and behavioural intervention (such as suggesting more appropriate dosing times for patients, educate them how to arrange the medications in pill-organizer boxes for easier use, educate them how set reminders for their medications different times) for the refugees could affect positively the adherent percentages and could result in more improvements. In addition, the study was conducted over three months only; it is suspected that with longer study duration more improvement in medication adherence would be resulted.’’

Reference: Not sure why reference list is duplicated. Please check. The duplicated references are deleted

Figure 2 &3: Is it possible to put the group baseline and follow-up scores on the graph too? These two figures (2 and 3) are for the mean changes in intervention and control groups for adherence and knowledge of medications, respectively.

Unfortunately, the scores cannot be inserted easily into the graphs, as the adherence questionnaire has 8 questions, and the knowledge of medication questionnaire has 5 questions. Inserting them into the figures will make it crowded and difficult to be understood. 

Please note that no changes were done on the manuscript in response to this comment.

Reviewer 3

Comment Response

Line 44: The word “chroesacnic” should be modified (chronic?) It is changed to ‘’chronic’’

Line 47: The objectives of the paper are “to evaluate the effect of the MMR service on adherence and on knowledge”. The identification of TRPs is not an objective of the paper. However, along the abstract, TRPs are described in methods, results (they are measured at baseline and at the end of the three months period) and cited as a conclusion. Conversely, in the body of the manuscript TRPs are hardly cited. In my opinion, authors should clarify the role of TRPs in this piece of research. The ultimate goals of the study were:

1- To investigate the baseline frequenct/type of TRPs among the Syrian refugees.

2- To investigate the impact of the MMR service on reducing the total number of TRPs

3- To assess the changes in TRPs’ frequencies and types between baseline and follow up 

4- To investigate the impact of applying MMR service on Refugees’ adherence to medication, knowledge of medication , quality of life, and anxiety scale between baseline and follow up. 

However, the results which are related to TRPs are published previously (Al Alawneh M, Nuaimi N, Basheti IA. Pharmacists in humanitarian crisis settings: Assessing the impact of pharmacist-delivered home medication management review service to Syrian refugees in Jordan. Res Social Adm Pharm. 2019;15(2):164-72. doi: 10.1016/j.sapharm.2018.04.008. Epub Apr 10). 

In addition, the results which are related to quality of life and anxiety are published previously too (Alawneh MA, Nuaimi N, Abu-Gharbieh E, Basheti IA. A randomized control trial assessing the effect of a pharmaceutical care service on Syrian refugees’ quality of life and anxiety. Pharmacy Practice. 2020;18(1))

- In this manuscript, the main objectives were to evaluate the effect of the MMR service on adherence and on knowledge of medications, with taking into consideration that the core of MMR service is the identyfication and correction of the TRPs. In another words, TRPs part of the study can not be deleted/ignored, as well as it can not be discussed in details in this manuscript. However, the above reference is reffered to many times through the manuscript.

The following sentence with the reference is added to Results section to make this issue clearer:

‘’The outcomes regarding the impact of the MMR on the TRPs’ type and frequency are reported elsewhere [Ref]’’

I hope this clarifies. 

Line 55: “Mediation” should be changed into “medication” Corrected

Line 63: TRPs are also included as a conclusion. So, in my opinion, they should be one of the objectives of the manuscript. The improvement in patients’ adherence and knowledge of medication is a result of delivering the MMR service and thus identifying and correction the TRPs. Knowing the positive impact of MMR service on TRPs, can explain the improvement in adherence and knowledge of medication which was found among the participants. For this reason, these TRPs are mentioned throughout the manuscript in many places. 

Adding the TRPs part to the objectives of this study is not possible, as it is not among the objectives of this study, and the TRPs-related results were published previously. However, TRPs part can be deleted from the Conclusion section, if it is preferred in your opinion.

Please note that no changes were done on the manuscript in response to this comment.

Very interesting description of the situation of Syrian refugees and the great amount of health problems they have to deal with, among other type of problems. However, for the purpose of this paper, I lack a further explanation on how these refugees manage to live there: do they live as a group in refugees camps, or do they locate or buy particular homes? Do they manage to find and have a job in Jordan or they live from charity? As in the paper the researcher goes to “homes”, in my opinion and taking into account I don’t have a proper knowledge of the real situation of this collective, this issue (socioeconomic situation of the refugees) should be clarified. The following paragraph is added to the Introduction section:

‘’To overcome this hypercritical situation, many camps were established in the country, such as Za’atari, Marjeeb al-Fahood, Cyber City and Al-Azraq camps. The UNHCR’s Home Visits Program Report has estimated that the largest number of Syrian refugees in Jordan are living outside these camps (84%); while the remaining are settled in urban and rural areas [Ref]. Primarily, Syrian refugees do not have a work permit to work in Jordan, and they are employed in the informal economy and they are not covered in the bounds of Jordanian labour law. However, approximately 51 % from the Syrian refugees who lives outside camps participates in the Jordanian labour market to afford an income for their families [Ref]. The unemployment rate for the total Syrian refugees in Jordan is considered high and it reaches 57 % [Ref]. [24]. Consequently, lower income, harder working conditions, tighter regulations, and lack of good work contracts are among the features of such ‘’work’’ for the refugees[Ref]. ‘’

The other issue with the introduction is that there is no reference to TRPs. Although they’re not included as an objective (I think perhaps they should) TRPs play a role in the paper, as they are included both in methods, and in conclusions. The following paragraph is added to the manuscript (Introduction section):

‘’The term TRP is widely used in the literature and is approved by major clinical areas across the world [Ref.]. The TRP can be defined as “an event or circumstance involving patient treatment that actually or potentially interferes with an optimum outcome for a specific patient”. It has been demonstrated in a previous longitudinal German study conducted between 2003 and 2007 that more than 5% of hospital visits are either caused or complicated by TRPs [Ref.]. Locally, a Jordanian study revealed that the identified TRPs in Jordanian outpatients with chronic diseases visiting community pharmacies are high [Ref.]’’

The following sentence is added too (Introduction section):

‘’number of TRPs among certain population’’

From the above I suggest to include TRPs evaluation as an objective. (In fact, in the ClinicalTrials web it is said that one of the objectives of the research is to evaluate the impact of MMR service) Evaluating the impact of MMR service on the frequency/type of TRPs was among the objectives of the whole RCT. However, in this manuscript the main objectives were to evaluate the effect of the MMR service on adherence and on knowledge of medications. TRP-related objectives -and consequently TRPs results- are published previously [Al Alawneh M, Nuaimi N, Basheti IA. Pharmacists in humanitarian crisis settings: Assessing the impact of pharmacist-delivered home medication management review service to Syrian refugees in Jordan. Res Social Adm Pharm. 2019;15(2):164-72. doi: 10.1016/j.sapharm.2018.04.008. Epub Apr 10]

Please note that no changes were done on the manuscript in response to this comment.

Line 129. It´s not clear for me the sentence: “The registration of this clinical study delayed, as it is not requested by Jordan). In my opinion, it should be clarified. This RCT has not been registered due to time restriction issue at the time of conduction, and due to the fact that the registration of the clinical trials is not mandatory in Jordan. However, the registration for this RCT was asked for by PLOS ONE journal to proceed with the submission. Consequently, the authors proceeded with the registration process, and the submission process after that.

The following word are added to the manuscript:

‘’ and due to time constraints’’

Line 138: It is no clear for me which is the “original study protocol” to which the authors refer to. The “original study protocol” should be clearly stated or described and supported by a bibliographic reference. The full ‘’original Study Protocol’’ was submitted as a separate file, as requested by the journal (PLOS ONE) in the beginning (it can be found among the files which were submitted by the authors previously). However, the concise Study protocol, which is for the current manuscript purposes, is included in this manuscript.

Please note that no changes were done on the manuscript in response to this comment.

Line 132-142: I wonder how the participants were approached and selected? There was any systematic design to identify possible participants or just those who went to the clinics in a particular period of time were contacted? The following health centres/clinics were visited, in order to meet and recruit the eligible Syrian refugees, as well as to arrange for their first home visits, for purpose of data collection. These centres/clinics were: Emirate Red Crescent clinics, Jordan health AID society centre, private clinic, social charity centre, governmental centre.

In another words, there was no systematic design to identify the participants. 

Please note that no changes were done on the manuscript in response to this comment.

Line 183: Adherence: The authors say that “adherence to medication was assessed by a developed and published questionnaire” (Reference #21). However, the questionnaire used by AbuRuz et al (21) is the well-known 8 items Morisky questionnaire. AbuRuz et al described the questionnaire they used as “a validated translation of the scale developed by Morisky et al”. In my opinion this should be clearly stated and referenced. The original name of the questionnaire and the suitable reference (Morisky 1986) is added to the manuscript (under Adherence to medication subheading), as follow:

which is called eight-item Morisky Medication Adherence Scale (MMAS-8)[Ref]

Line 198: Knowledge. Again, the authors refer to the paper by AbuRuz (21) while explaining the methodology related to the measurement of knowledge about chronic medication. Surprisingly, they say that they used a “validated and published questionnaire” but in the cited paper (21) no questionnaire is used. In fact, measuring knowledge wasn´t an objective of the AbuRuz´ paper. The right reference for AbuRuz (AbuRuz et al. 2006), is added to the Method section (under Knowledge of chronic medication section).

In AbuRuz et al. (2006), the knowledge of medication questionnaire was used (Please see Table 1 in AbuRuz study). 

A reference of a previous study which also used the Knowledge of medication questionnaire is added too (Basheti et al. 2016) to the Method section (under Knowledge of chronic medication section).

In line 267, the authors make a reference to the “five domains of the knowledge about drug therapy´ questionnaire” which is shown in table 4. That affirmation should be supported by a literature reference different from AbuRuz´. The right reference for AbuRuz (AbuRuz et al. 2006), is added to the Method section (under Knowledge of chronic medication section).

In addition, if a “validated questionnaire had been used” any further modification should be well explained and tested (at least piloted) in order to not to get their properties loosen. I suggest the authors should rephrase this section explaining the procedure and their bibliographic support. The questionnaire was modified by the research team to include close ended questions instead of open-ended question. This was done for the purpose of facilitating the reading, understanding, and choosing the answers by the Syrian refugees, who some of them might not have the adequate literacy to fill complicated questionnaire.

Please note that no changes were done on the manuscript in response to this comment.

Line 276: Although TRPs are not included as an objective and they´re not described in the results section, the authors states that: “…decreasing the number of TRPs they suffer from”. This assertion should come from previously described results. So, it should be rephrased. The reference which indicated the previously published results of decreasing the TRPs among the refugees was added at the end of the sentence (Please see the modified manuscript Word version)

From line 300 to line 314 there is a comparison of the adherence results with those of four other papers (15, 30, 31, 32, 33), comparison that in my opinion makes the discussion too long. I recommend shortening this paragraph. ‘’Brummel & Carlson conducted a study in 2016 revealing that the comprehensive medication management (CMM) service among USA participants resulted in significant improvement in their medication adherence . Basheti et al. conducted a study in 2016 as well, showing that the MMR service resulted in a significant reduction of non-adherence scores among the intervention group participants’’ was shortened to be ‘’ two studies were conducted in 2016 and revealed that the MMR and the comprehensive medication management (CMM) services -respectively- resulted in significant improvement in medication adherence among the studies’ samples’’. Other sentences within the indicated paragraph were shortened too (please see the modified Word version). 

On the other hand, the authors found that at baseline, adherence ranged between 13.2% in the IG and the 11.3% in CG (p value = 0.540). After the follow up, adherence improved until the 32.1%. These figures mean that at baseline non-adherence rates among Syrian refugees were higher than 80% (IG: 86.8%; CG:88.7%) and that after the intervention a 67.9% of non-adherence was achieved in the IG. I think these figures should be commented in the Discussion because they show really high rates of non-adherence at baseline (and even at the end of the study, being it very successful). The following paragraph is added to the Discussion section to address this comment:

‘’The adherence percentages among both groups at base line were really low, and this can be explained-in part- by the nature of the study sample and the harsh life conditions that they live in, which lack the fundamental elements of normal life. Although the percentage of adherent patients form the intervention group has been improved to reach 32%, yet this percentage is not sufficient for proper health management. However, further follow up visits/calls, which include educational and behavioural intervention (such as suggesting more appropriate dosing times for patients, educate them how to arrange the medications in pill-organizer boxes for easier use, educate them how set reminders for their medications different times) for the refugees could affect positively the adherent percentages and could result in more improvements [Ref]. In addition, the study was conducted over three months only; it is suspected that with longer study duration more improvement in medication adherence would be resulted.’’

Another issue that, in my opinion, deserves a comment in the discussion is that only a 14.5% of the non-adherent patients (n=15; table 2) were really unintentional. I could not understand this comment. The mentioned percentage (14.5) and the number of patients (n=15) is not found on the indicated table. Please clarify. 

Line 347: Again, there is a reference to TRPs which haven´t been described in the results section. TRPs results were published previously. A reference is added in the Conclusion section to refer to. 

Literature references are repeated and have different numbers. References should be revised.

Line 363 – 463 References 1 – 36

Line 555 – 658 References 1 – 37

Line 659 – 760 References 1 – 37

Line 761 – 864 References 1 – 37

Line 865 – 968 References 1– 37

Line 969 – 1057 References 1 – 34 The references were revised, and the duplication were deleted. 

Reviewer 4

Comment Response

Major issue 1. 

Qualitative answers are transformed into numeric values. Mapping the support of an ordered factor to the support of a numeric variable is in general not recommended, as it relies on a arbitrary choice. It is also not necessary, because generalized linear models (GLM) for ordered variables are available for analysis (also in SPSS, which is the software used in this paper). A generalized linear model (SPSS) is used to analyse the ordered variables in both questionnaires, as requested. Some changes were done in the Method and Results sections accordingly. 

Major issue 2. 

The response variable are obtained by summing scores derived from different questions. Summing heterogeneous scores that reflect different dimensions of adherence and knowledge make the interpretation of the response variable rather difficult: for instance, two subjects with different patterns of answers could obtain the same score. This is tolerable when the items are exchangeable. Is it the case here? If it is not, then the response variable should be treated as a multivariate response. An item response model is the state-of-the art method to analyze multiple items in a questionnaire.

I don't know whether SPSS includes these models, but they are available in many software packages such as R and SAS. A generalized linear model is used to analyse the multiple items in both questionnaires, as requested. Changes were done accordingly in the Methods and Results sections.

Major issue 3. 

Variable adherence is not a continuous variable: it is a discrete variable with range 0-32. Perhaps (not clear from text, see specific point 2) variable knowledge is discrete too. The t-test is therefore not fully appropriate in this context. Generalized linear mixed models (available in SPSS, which is the software used in the paper) provide the state-of-art strategy to compare discrete responses in a longitudinal setting. Adherence scores – as well as the knowledge of medications scores- are analysed as a likert scale out of 5 (they are considered as discrete variable instead of continuous variable); it is more appropriate for the type of the discrete data. Changes in the Methods and Results were done accordingly. The following sentense is added to Statistical analysis subheading:

‘’Likert scale scores (for adherence and knowledge of medication questionnaire) were analysed using the generalized linear model.’’

lines 183 - 189. Only 5 questions are described, while the questionnaire should include 8 questions: please clarify. It is clarified in the Method section (under Adherence to medication subheading), as requested.

Lines 204-205. The continuous scale out of 5 is not described. What are the possible scores? The integers 1 to 5? Or any real number between 1 and 5? Please clarify. The following paragraph is added to the indicated section in the manuscript:

‘’The measurement scale used in this questionnaire was scored at 0 (I know this very well about all medications which I use), 1 (I know this to some extent about all medications which I use), 2 (I know this to some of my medications only), 3 (I do not know) and 4 (I do not know at all). Hence, knowledge of chronic medication was analysed as a continuous scale out of 20.’’

---

## [Decision Letter · Decision Letter 1]

20 Sep 2021

PONE-D-20-26688R1Improving Syrian refugees’ knowledge of medications and adherence following a randomized control trial assessing the effect of a medication management review servicePLOS ONE

Dear Dr. Basheti,

Thank you for submitting your manuscript to PLOS ONE. After careful consideration, we feel that it has merit but does not fully meet PLOS ONE’s publication criteria as it currently stands. Therefore, we invite you to submit a revised version of the manuscript that addresses the points raised during the review process.

 The reviewers have raised a number of outstanding concerns that need to be carefully addressed in a revision to your manuscript.

We look forward to receiving your revised manuscript.

Kind regards,

Jamie Males

Staff Editor

PLOS ONE

Reviewers' comments:

Reviewer's Responses to Questions

**Comments to the Author**

1. If the authors have adequately addressed your comments raised in a previous round of review and you feel that this manuscript is now acceptable for publication, you may indicate that here to bypass the “Comments to the Author” section, enter your conflict of interest statement in the “Confidential to Editor” section, and submit your "Accept" recommendation.

Reviewer #1: All comments have been addressed

Reviewer #3: All comments have been addressed

Reviewer #4: (No Response)

2. Is the manuscript technically sound, and do the data support the conclusions?

Reviewer #1: Yes

Reviewer #3: Partly

Reviewer #4: No

3. Has the statistical analysis been performed appropriately and rigorously? 

Reviewer #1: Yes

Reviewer #3: Yes

Reviewer #4: No

4. Have the authors made all data underlying the findings in their manuscript fully available?

Reviewer #1: Yes

Reviewer #3: Yes

Reviewer #4: No

5. Is the manuscript presented in an intelligible fashion and written in standard English?

Reviewer #1: Yes

Reviewer #3: Yes

Reviewer #4: Yes

6. Review Comments to the Author

Reviewer #1: Thank you for addressing all comments. I have just one concern regarding the sample size. Since the study is underpowered, the results obtained might not be accurate. I will leave it up to the editor in chief to make the decision about accepting the paper or not.

Good luck.

Reviewer #3: Dear editor,

I have read the comments by the authors to my review recommendations (and to the rest of the reviewers´) and in my opinion the paper has improved a lot.

I agree with most of the comments of the authors, even though in some occasions they haven’t include any changes on the manuscript. However, there is one aspect I think they should re-address, which is using a really an ad-hoc questionnaire as a formerly-validated one.

To my comment: “In addition, if a “validated questionnaire had been used” any further modification should be well explained and tested (at least piloted) in order to not to get their properties loosen. I suggest the authors should rephrase this section explaining the procedure and their bibliographic support.

The authors‘ answer is: The questionnaire was modified by the research team to include close ended questions instead of open-ended question. This was done for the purpose of facilitating the reading, understanding, and choosing the answers by the Syrian refugees, who some of them might not have the adequate literacy to fill complicated questionnaire.

Please note that no changes were done on the manuscript in response to this comment.

I can understand the purpose of the authors in doing that, but I can´t agree with them and so, with the redaction of the paragraph. In my opinion a “validated questionnaire” can´t be modified in any sense and so, the structure of the questions going from open-ended questions to close-ended questions, can´t be changed. This should be specified in the paper, perhaps including that “an ad hoc questionnaire was used based on a previously validated one, to help understanding to the respondents”.

Apart from that issue, I think the paper can be published.

Reviewer #4: PONE-D-20-26688R1: statistical review

The authors have not addressed the major concerns I raised in my previous review. Specifically, they say that they use a generalized linear model to address all the issues, without explaining what kind of generalized linear model has been chosen for each issue. I therefore copy and paste below the three issues that I raised in my previous review. Each issue must be addressed by describing a specific, well-motivated model.

MAJOR ISSUES

1. Qualitative answers are transformed into numeric values. Mapping the support of an ordered factor to the support of a numeric variable is in general not recommended, as it relies on a arbitrary choice. It is also not necessary, because generalized linear models (GLM) for ordered variables are available for analysis (also in SPSS, which is the software used in this paper).

2. The response variable are obtained by summing scores derived from different questions. Summing heterogeneous scores that reflect different dimensions of adherence and knowledge make the interpretation of the response variable rather difficult: for instance, two subjects with different patterns of answers could obtain the same score. This is tolerable when the items are exchangeable. Is it the case here? If it is not, then the response variable should be treated as a multivariate response. An item response model is the state-of-the art method to analyze multiple items in a questionnaire.

I don't know whether SPSS includes these models, but they are available in many software packages such as R and SAS.

3. Variable adherence is not a continuous variable: it is a discrete variable with range 0-32. Perhaps (not clear from text, see specific point 2) variable knowledge is discrete too. The t-test is therefore not fully appropriate in this context. Generalized linear mixed models (available in SPSS, which is the software used in the paper) provide the state-of-art strategy to compare discrete responses in a longitudinal setting.

7. PLOS authors have the option to publish the peer review history of their article (what does this mean?). If published, this will include your full peer review and any attached files.

Reviewer #1: No

Reviewer #3: No

Reviewer #4: No

---

## [Author Response · Author response to Decision Letter 1]

4 Nov 2021

We would like to thank you very much for the constructive comments you provided on the manuscript. The authors highly appreciate the comments and strongly believe that the recommendations would positively impact the quality and the merit of the paper. We have revised the manuscript and incorporated appropriate changes to address the issues you have raised, and we provide a response letter detailing to each of your comments.

---

## [Decision Letter · Decision Letter 2]

17 Jan 2022

PONE-D-20-26688R2Improving Syrian refugees’ knowledge of medications and adherence following a randomized control trial assessing the effect of a medication management review servicePLOS ONE

Dear Dr. Basheti,

Thank you for submitting your manuscript to PLOS ONE. After careful consideration, we feel that it has merit but does not fully meet PLOS ONE’s publication criteria as it currently stands. Therefore, we invite you to submit a revised version of the manuscript that addresses the points raised during the review process.

The manuscript has been evaluated by three reviewers, and their comments are available below.

The reviewers comments have mostly been addressed, however there are some concerns that need attention. They request some alterations to the Abstract. In addition, the concerns about the sample size should be further addressed in the Limitations and the conclusions should be interpreted with this in mind.

Could you please revise the manuscript to carefully address the concerns raised?

We look forward to receiving your revised manuscript.

Kind regards,

Jamie Royle

Associate Editor

PLOS ONE

Reviewers' comments:

Reviewer's Responses to Questions

**Comments to the Author**

1. If the authors have adequately addressed your comments raised in a previous round of review and you feel that this manuscript is now acceptable for publication, you may indicate that here to bypass the “Comments to the Author” section, enter your conflict of interest statement in the “Confidential to Editor” section, and submit your "Accept" recommendation.

Reviewer #1: All comments have been addressed

Reviewer #3: (No Response)

Reviewer #4: All comments have been addressed

2. Is the manuscript technically sound, and do the data support the conclusions?

Reviewer #1: Yes

Reviewer #3: Yes

Reviewer #4: (No Response)

3. Has the statistical analysis been performed appropriately and rigorously? 

Reviewer #1: Yes

Reviewer #3: Yes

Reviewer #4: (No Response)

4. Have the authors made all data underlying the findings in their manuscript fully available?

Reviewer #1: Yes

Reviewer #3: Yes

Reviewer #4: (No Response)

5. Is the manuscript presented in an intelligible fashion and written in standard English?

Reviewer #1: Yes

Reviewer #3: Yes

Reviewer #4: (No Response)

6. Review Comments to the Author

Reviewer #1: (No Response)

Reviewer #3: Thanks for allowing me to review the new version of the manuscript on Improving Syrian refugee’s adherence and knowledge by a pharmacist intervention.

I find it very improved and I can only add a couple of suggestions:

(a) In my opinion, the key word “SDG 3” should be removed.

(b) In the Results section it is said that “the outcomes regarding the impact of the MMR on the TRPs’ type and frequency are reported elsewhere [28]” (Line 272). However, in the abstract some data on these results are included (At follow-up, a significant decrease in the number of TRPs for refugees in the intervention group was found ((from 600 to 182, P <0.001), but not for control group (number stayed at 541 TRPs, P= 0.116) (Line 60). I think that this information should, also, be removed from the Abstract

Reviewer #4: (No Response)

7. PLOS authors have the option to publish the peer review history of their article (what does this mean?). If published, this will include your full peer review and any attached files.

Reviewer #1: No

Reviewer #3: **Yes: **miguel angel gastelurrutia

Reviewer #4: No

---

## [Author Response · Author response to Decision Letter 2]

25 Jan 2022

We would like to thank you very much for the constructive comments you provided on the manuscript. The authors highly appreciate the comments and strongly believe that the recommendations would positively impact the quality and the merit of the paper. We have revised the manuscript and incorporated appropriate changes to address the issues you have raised.

---

## [Decision Letter · Decision Letter 3]

30 Mar 2022

PONE-D-20-26688R3Improving Syrian refugees’ knowledge of medications and adherence following a randomized control trial assessing the effect of a medication management review servicePLOS ONE

Dear Dr. Basheti,

Thank you for submitting your manuscript to PLOS ONE. After careful consideration, we feel that it has merit but does not fully meet PLOS ONE’s publication criteria as it currently stands. Therefore, we invite you to submit a revised version of the manuscript that addresses the points raised during the review process.

The manuscript has been evaluated by several reviewers, and their comments are available below.<o:p></o:p>

The reviewers have not raised any major concerns concerning your manuscript. However during our in house checks we noticed you have used the 8 point Morisky Medication Adherence Scale. The authors of this scale patented the scale and it can only be used under license. Notably also we note you have included in your text several of the questions from the scale This text is copyrighted and can not be published. Before we can go forward with your manuscript these copyright issues need to be resolved. <o:p></o:p>

We look forward to receiving your revised manuscript.

Kind regards,

Thomas Phillips, PhD

Associate Editor

PLOS ONE

Reviewers' comments:

Reviewer's Responses to Questions

**Comments to the Author**

1. If the authors have adequately addressed your comments raised in a previous round of review and you feel that this manuscript is now acceptable for publication, you may indicate that here to bypass the “Comments to the Author” section, enter your conflict of interest statement in the “Confidential to Editor” section, and submit your "Accept" recommendation.

Reviewer #1: All comments have been addressed

Reviewer #3: All comments have been addressed

Reviewer #4: All comments have been addressed

2. Is the manuscript technically sound, and do the data support the conclusions?

Reviewer #1: Yes

Reviewer #3: Yes

Reviewer #4: (No Response)

3. Has the statistical analysis been performed appropriately and rigorously? 

Reviewer #1: Yes

Reviewer #3: Yes

Reviewer #4: (No Response)

4. Have the authors made all data underlying the findings in their manuscript fully available?

Reviewer #1: Yes

Reviewer #3: Yes

Reviewer #4: (No Response)

5. Is the manuscript presented in an intelligible fashion and written in standard English?

Reviewer #1: Yes

Reviewer #3: Yes

Reviewer #4: (No Response)

6. Review Comments to the Author

Reviewer #1: The authors have addressed my comments. Thank you for your efforts and congratulations for the publication.

The paper is up to my expectations.

Reviewer #3: The paper now has been improved and as it approaches a no very common issue can be interesting for many people. Thanks for having addressed my comments. In my opinion the paper is ready to be published.

Reviewer #4: (No Response)

7. PLOS authors have the option to publish the peer review history of their article (what does this mean?). If published, this will include your full peer review and any attached files.

Reviewer #1: No

Reviewer #3: No

Reviewer #4: No

---

## [Author Response · Author response to Decision Letter 3]

10 Apr 2022

The authors have been responded to the comment which raised by the editor.

---

## [Decision Letter · Decision Letter 4]

28 Jul 2022

PONE-D-20-26688R4Improving Syrian refugees’ knowledge of medications and adherence following a randomized control trial assessing the effect of a medication management review servicePLOS ONE

Dear Dr. Basheti,

Thank you for submitting your manuscript to PLOS ONE. After careful consideration, we feel that it has merit but does not fully meet PLOS ONE’s publication criteria as it currently stands. Therefore, we invite you to submit a revised version of the manuscript that addresses the points raised during the review process. Two minor updates are required before publication:

A) Please upload a completed CONSORT checklist as a supporting information file. Blank copies of this document and information regarding CONSORT can be found via the following link: http://www.consort-statement.org/. Please ensure that you update your manuscript to include the information requested in the checklist. B) The information in the registry entry suggests that your trial was registered after patient recruitment began. PLOS ONE strongly encourages authors to register all trials before recruiting the first participant in a study. As per the journal’s editorial policy, please include in the Methods section of your paper: 1) your reasons for your delay in registering this study (after enrolment of participants started); 2) confirmation that all related trials are registered by stating: “The authors confirm that all ongoing and related trials for this drug/intervention are registered”. C) Optional: I suggest that the current flow chart is replaced with a copy using the CONSORT template. Blank copies of this document and information regarding CONSORT can be found via the following link: http://www.consort-statement.org/.

We look forward to receiving your revised manuscript.

Kind regards,

George Vousden

Deputy Editor in Chief

PLOS ONE

Journal Requirements:

Reviewers' comments:

Reviewer's Responses to Questions

**Comments to the Author**

1. If the authors have adequately addressed your comments raised in a previous round of review and you feel that this manuscript is now acceptable for publication, you may indicate that here to bypass the “Comments to the Author” section, enter your conflict of interest statement in the “Confidential to Editor” section, and submit your "Accept" recommendation.

Reviewer #1: All comments have been addressed

Reviewer #4: All comments have been addressed

2. Is the manuscript technically sound, and do the data support the conclusions?

Reviewer #1: Yes

Reviewer #4: (No Response)

3. Has the statistical analysis been performed appropriately and rigorously? 

Reviewer #1: Yes

Reviewer #4: (No Response)

4. Have the authors made all data underlying the findings in their manuscript fully available?

Reviewer #1: Yes

Reviewer #4: (No Response)

5. Is the manuscript presented in an intelligible fashion and written in standard English?

Reviewer #1: Yes

Reviewer #4: (No Response)

6. Review Comments to the Author

Reviewer #1: Dear authors,

I have no more comments. I am satisfied with the revision. The paper can be accepted for publication.

Reviewer #4: (No Response)

7. PLOS authors have the option to publish the peer review history of their article (what does this mean?). If published, this will include your full peer review and any attached files.

Reviewer #1: No

Reviewer #4: No

---

## [Author Response · Author response to Decision Letter 4]

14 Aug 2022

A file named ''Response to Reviewers'' containing the comments and the authors' responses is uploaded, for your kind consideration.

---

## [Editor Report · Decision Letter 5]

5 Oct 2022

Improving Syrian refugees’ knowledge of medications and adherence following a randomized control trial assessing the effect of a medication management review service

PONE-D-20-26688R5

Dear Dr. Basheti,

We’re pleased to inform you that your manuscript has been judged scientifically suitable for publication and will be formally accepted for publication once it meets all outstanding technical requirements.

Kind regards,

George Vousden

Staff Editor

PLOS ONE
---

## [Editor Report · Acceptance letter]

7 Oct 2022

PONE-D-20-26688R5 

Improving Syrian refugees’ knowledge of medications and adherence following a randomized control trial assessing the effect of a medication management review service 

Dear Dr. Basheti:

I'm pleased to inform you that your manuscript has been deemed suitable for publication in PLOS ONE. Congratulations! Your manuscript is now with our production department. 

Kind regards, 

on behalf of

Dr. George Vousden 

Staff Editor

PLOS ONE